# Racial and Ethnic Disparities in Genomic Healthcare Utilization, Patient Activation, and Intrafamilial Communication of Risk among Females Tested for *BRCA* Variants: A Mixed Methods Study

**DOI:** 10.3390/genes14071450

**Published:** 2023-07-15

**Authors:** Sharlene Hesse-Biber, Memnun Seven, Hannah Shea, Madeline Heaney, Andrew A. Dwyer

**Affiliations:** 1Department of Sociology, Boston College, Chestnut Hill, MA 02467, USA; hesse@bc.edu (S.H.-B.); hcshea99@gmail.com (H.S.); heaneymc@bc.edu (M.H.); 2Elaine Marieb College of Nursing, University of Massachusetts Amherst, Amherst, MA 01003, USA; 3William F. Connell School of Nursing, Boston College, Chestnut Hill, MA 02467, USA; andrew.dwyer@bc.edu; 4Harvard Center for Reproductive Medicine, Massachusetts General Hospital, Boston, MA 02114, USA

**Keywords:** *BRCA* mutation, genomic healthcare, intrafamilial communication of risk, disparities

## Abstract

**Simple Summary:**

Decision-making on intrafamilial communication of Breast Cancer gene (*BRCA)* risk and subsequent cascade screening is complex and affected by individual, interpersonal, and healthcare system-related factors. There is a paucity of evidence on factors affecting intrafamilial risk communication, especially among ethnically and racially diverse individuals tested for *BRCA* variants. A deep understanding of multi-level factors affecting communication is needed to reap the full potential of cascade screening for improving health outcomes. This study provides findings to guide theory-driven, multi-level interventions to enhance the utilization of genomic healthcare for diverse people.

**Abstract:**

This study aimed to gain a deeper understanding of genomic healthcare utilization, patient activation, and intrafamilial risk communication among racially and ethnically diverse individuals tested for *BRCA* variants. We employed an explanatory, sequential, mixed-methods study guided by the Theory of Planned Behavior. Participants completed an online survey, including sociodemographic, medical history, and several validated instruments. A subset of participants participated in in-depth, semi-structured interviews. A total of 242 women were included in the quantitative analyses. The majority of survey participants identified as non-Hispanic white (NHW) (n = 197, 81.4%) while 45/242 (18.5%) identified as black, Indigenous, and people of color (BIPOC). The NHW participants were more likely to communicate genetic test results with healthcare providers, family, and friends than BIPOC participants (*p <* 0.05). BIPOC participants had lower satisfaction with testing decisions and significantly higher ratings of personal discrimination, fatalism, resilience, uncertainty, and lower patient activation scores (*p <* 0.05). Participants with higher education, greater satisfaction with testing decisions, and lower resilience are more likely to communicate *BRCA* test results with family members through the mediating effect of patient activation. Bridging disparities to ensure that genomic healthcare benefits all people may demand theory-driven, multi-level interventions targeting the individual, interpersonal, and healthcare system levels.

## 1. Introduction

Cancer incidence and mortality vary across racial and ethnic populations and geographic regions in the United States (U.S.) [1]. Compared to non-Hispanic White (NHW) people, Black, Indigenous, and People of Color (BIPOC), as well as other historically underserved populations, continue to share a disproportionate burden for certain cancers [1]. Breast cancer is the most common cancer type among American women. Despite similar cancer incidence rates, black women have a 41% higher breast cancer death rate compared to NHW women [2]. Inherited pathogenic germline variants underlie approximately 5–10% of all breast cancer cases—most commonly in *BRCA1* and *BRCA2* (*BRCA*) genes [3]. Individuals harboring pathogenic *BRCA* variants have a 69–72% lifetime risk of breast cancer and a 17–44% risk of ovarian cancer [4], compared with lifetime risks of 12.9% and 1.1% in the general population, respectively [5]. Clinical guidelines have been established to assist clinicians in identifying people at high risk of cancer who may benefit from further genetic counseling and testing (i.e., cascade carrier screening) to improve cancer outcomes [6].

Cascade carrier screening is a valuable tool for early cancer detection and intervention among family members (i.e., blood relatives) who are asymptomatic or undiagnosed [7,8]. Regardless of the test findings, genetic testing can benefit individuals undergoing testing and their families. A positive result can inform health promotion, enhanced screening, and/or risk-reducing treatment/surgery, reducing morbidity and mortality [6,9,10]. On the other hand, a negative result test (or finding of a variant of uncertain significance) may reduce concern for cancer and subsequent risk-reducing measures [10]. Importantly, cascade screening is only possible when an individual who tests positive communicates their test results to at-risk blood relatives [7]. However, learning one’s *BRCA*+ status can be accompanied by emotional responses, including shock, denial, uncertainty, guilt, or even regret, that may pose barriers to intrafamilial communication of risk [11,12]. As such, there appears to be a need for *BRCA*+ individuals to be ‘activated’ to effectively communicate results to potentially at-risk blood relatives and initiate cascade screening. Patient activation refers to an individual’s knowledge, skill, and confidence in managing a health condition, collaborating with healthcare providers, and accessing appropriate care [13]. Activation has important links to one’s coping response—a critical factor that precedes intrafamilial communication of risk enabling subsequent cascade screening [14].

Despite the effectiveness of genomic healthcare (i.e., genetic testing and cascade screening) for improving outcomes, evidence suggests such approaches are underutilized by BIPOC and medically underserved groups [9,15]. Recent data indicate that Black women with ovarian cancer are significantly less likely to receive genetic testing than their NHW counterparts [1]. Further, data suggest that Black women are less willing than NHW counterparts to disclose test results to family members—a crucial step enabling cascade screening [16,17]. Similarly, studies indicate Black and Hispanic individuals are less likely to discuss *BRCA* testing with their providers, undergo genetic testing, and receive risk-reducing interventions [3,6,16]. These data indicate that not all communities benefit equally from genomic healthcare and the resulting potential for improved cancer outcomes (i.e., morbidity/mortality). A deeper understanding of genomic healthcare, patient activation, and intrafamilial communication of risk is essential to reap the full potential of cascade screening for improving outcomes. Therefore, this study aimed to gain a deeper understanding of genomic healthcare utilization, patient activation, and intrafamilial risk communication among racially diverse females who tested for *BRCA* mutations.

## 2. Materials and Methods

This study employed an explanatory sequential, mixed-methods study to examine genomic healthcare utilization, health activation, and intrafamilial communication of *BRCA* risk among ethnically and racially diverse individuals. The Boston College IRB approved this research study (protocol #16.109.01), and all participants provided opt-in electronic informed consent before study participation. Findings are reported according to Strengthening the Reporting of Observational studies in Epidemiology (STROBE) [18].

### 2.1. Participants and Procedures

The study sample included English- or Spanish-speaking adults (18+ years.) who underwent genetic testing for *BRCA1/2* (i.e., hereditary breast and ovarian cancer, HBOC). Participants were recruited between February and October 2022. We used targeted strategies to enhance the purposive recruitment of diverse individuals by specifically targeting BIPOC individuals. We employed community-engaged outreach in collaboration with patient support organizations (Facing Our Risk of Cancer Empowered [FORCE], The Black Breast Cancer Alliance [TOUCH], SurviveHer, and AfroPink). Following opt-in informed consent, participants completed an online survey (quantitative), including sociodemographic information, medical history, and several validated instruments (Figure 1). A subset of participants was invited to participate in in-depth, semi-structured qualitative interviews (45–90 min. in duration).

### 2.2. Theoretical Framework

This work was guided by the Theory of Planned Behavior (TPB) [19] to better understand patient activation and familial risk communication of risk. The TPB posits that intention precedes action (i.e., patient activation precedes intrafamilial risk communication). Intentions are shaped by attitudes (i.e., positive or negative perceptions of an action/behavior), subjective norms (i.e., expectations of family, friends, or healthcare professionals), and perceived behavioral control (i.e., sense of agency and perceptions that the action is up to the individual). The mediators of intention (attitudes, subjective norms, and perceived behavioral control) are influenced by beliefs, values, and an individual’s past experiences [19]. We used the TPB as a framework to map key study variables and to inform the subsequent development of targeted interventions to enhance patient activation (intention) and intrafamilial communication of risk (behavior) (Figure 2). When considering targeted interventions, we also utilized an ecological perspective to categorize interventions at the individual, interpersonal (family), and healthcare system levels. Theory-driven interventions are recommended by the United Kingdom’s Medical Research Council, and robust evidence supports that theory-informed approaches are more successful than “a-theoretical” complex interventions [20].

### 2.3. Quantitative Survey

For the online quantitative survey, participants provided socio-demographic data (i.e., age, race, education, marital status, and a validated single-item health literacy question [21], and reproductive information (i.e., childbearing status and number of biological/adopted children). In addition, we collected data on personal and family history of cancer, *BRCA* genetic counseling and testing, reasons for testing, risk management strategies (e.g., enhanced screening and risk-reducing surgery), and communication of *BRCA* risk with family members (Interfamilial communication) and healthcare providers. To assess experiences with healthcare, we used the Genetic Counseling Satisfaction Scale (GCSS), a 6-item Likert-type scale assessing participant satisfaction with the process and content of genetic counseling [22]. We measured patient satisfaction using the Satisfaction with Decision Scale (SWD), a 6-item Likert-type scale with higher scores indicating greater satisfaction [23]. We also employed the Decision Regret Scale (DRS), a 5-item Likert-type scale assessing remorse related to a decision (i.e., genetic testing) [24]. To assess experiences of discrimination, we used three validated instruments to assess and triangulate perceptions and experiences of discrimination in healthcare. The single-item Personal Discrimination Survey has been used widely to examine perceptions of discrimination in health services research. The multi-item Personal Discrimination Survey is a 7-item instrument wherein responses are dichotomized (never vs. ever) and tabulated into an overall score of perceived discrimination [25]. The General Racism in the Healthcare System instrument uses 5-point Likert-types ratings (strongly disagree to strongly agree) to assess the extent to which individuals agree with four statements regarding racial discrimination in healthcare settings, with higher scores indicating greater perceived racism in healthcare [25]. Importance of religion was measured by asking participants, ‘How important is religion in your life?’ on a visual scale, with 0 being “not important at all” and 10 being “extremely important”.

We used several validated measures to assess coping responses. The Multidimensional Impact of Cancer Risk Assessment (MICRA) is a 21-item scale assessing the impact of results disclosure after genetic testing. The instrument has three subscales: distress, uncertainty, and positive experiences, with scores ranging from 0–100. The MICRA is widely used in cancer research and provides a targeted assessment of concerns and psychosocial issues associated with genetic testing for cancer risk [26]. The 20-item Fatalism Scale conceptualizes the construct of fatalism as a set of health beliefs across three dimensions: predetermination (the belief that all events are arranged/decided before happening), luck (attribution of one’s health/life events as a matter of chance), and pessimism (tendency to believe that the worse will occur). The total score provides insight into the overall role of fatalism in health behavior [27]. The 6-item Brief Resilience Scale (BRS) assesses an individual’s ability to ‘bounce back’ or recover from a stressful life experience. Items are framed positively/negatively (with reverse scoring, respectively), and higher scores indicate a greater ability to recover from stressful events or adversity [28]. Last, we measure patient activation using the 13-item Patient Activation Measure (PAM) that assesses patient knowledge, skill, and confidence in self-managing a health condition, collaborating with healthcare providers, and accessing appropriate care. Scores range from 0–100, with higher scores indicating greater knowledge, skill, and confidence in self-management behaviors and behavior changes [13].

### 2.4. Qualitative Interviews

In-depth, semi-structured interviews were used to contextualize and elaborate on quantitative survey findings and examine the psychosocial impact of *BRCA* testing, coping response to *BRCA* status, intrafamilial communication of *BRCA* risk, and subsequent cascade screening. Each online interview began by reviewing sociodemographic information followed by questions about the personal history of cancer and genetic testing, healthcare experiences during the testing (i.e., decision-making process), communication of genetic test results with blood relatives, perceived effect of sociodemographic on their experiences, as well as physical, emotional, and psychological reactions to genetic test results. Interviews lasted 45 to 90 min. Memos were recorded for each interview, and audio-recorded interviews were transcribed verbatim. Participants received a USD 40 gift certificate as remuneration. Interviewees were allowed to review their transcribed interviews and provide edits/clarifications to their responses. No participants opted to review their interview transcript. Qualitative results were depicted using a “joint display” [29]. The joint display enables the direct linkage of significant quantitative survey findings with qualitative findings emerging from the template analysis, thereby providing social context and insights.

### 2.5. Analyses

The quantitative data were analyzed using the IBM SPSS Statistics 28 program [30,31]. Results are reported descriptively, and Kolmogorov–Smirnov tests were used to evaluate the data distribution. Fisher’s exact Tests and Mann–Whitney-U Tests were employed for non-normally distributed data. Fisher’s exact Tests for the bivariate comparison among BIPOC and NHW were used to consider the potential small sample size in an individual cell. Drawing on the guiding TBP framework, structural equation modeling (SEM) was conducted to assess multivariate factors predicting health activation, intrafamilial communication of results, and individual health behaviors (i.e., enhanced screening, risk-reducing surgery). For the SEM model, the sample calculation is based on the total sample size rather than specific variable distribution. The general rule is to either have a sample size of 100–200 or 5–10 observations per estimated parameter [32]. A *p* value <0.05 was considered statistically significant.

For qualitative interviews, transcripts were analyzed using template analysis [33,34,35]. In brief, template analysis involves independent iterative coding (S.H.-B., H.S., and M.H.) using an a priori coding template of themes. During coding, “sub-themes” may emerge representing dimensions of a template “theme”. For this study, the a priori themes were drawn from the TPB. Once data saturation is met (i.e., when no new themes or sub-themes emerge from interview transcripts), two independent investigators (S.H.-B. and H.S.) mapped salient findings to the TPB to inform the subsequent design of targeted interventions to enhance patient activation and intrafamilial communication of *BRCA* risk for racially and ethnically diverse populations.

## 3. Results

### 3.1. Participant Characteristics

Overall, 472 participants accessed the online survey. After excluding those with male sex assigned at birth (n = 109), invalid entries (n = 77), and those who had not undergone *BRCA* testing (n = 44), a total of 242 valid entries were included for analysis. The sociodemographic characteristics of survey respondents are depicted in Table 1. Respondents ranged in age from 20–76 years. Based on self-reported race and ethnicity, the majority of survey participants identified as NHW (n = 197, 81.4%)—while 45/242 (18.5%) identified as BIPOC (Black/African American: n= 29 [12.0%], American Indian/Alaska Native: n = 4 [1.7%], Native Hawaiian/Pacific Islander: n = 3 [1.2%], Asian/Asian-American: n = 8 [3.3%], mixed/multiracial: n = 1 [0.4%]). The NHW and BIPOC participants were similar regarding gender identity, marital status, income, and having children. We observed statistical differences (*p <* 0.05) concerning age, education, health literacy, number of children, and importance of religion between NHW and BIPOC participants.

### 3.2. Participant Medical Information and Genetic Testing

Information on medical care and genetic testing is shown in Table 2. The BIPOC and NHW participants were similar regarding rates of harboring a pathogenetic *BRCA* variant, inheritance (i.e., maternal vs. paternal), and the primary reason for testing. Personal history of cancer and family history of breast cancer (among first-degree male relatives) were significantly higher among BIPOC participants. However, a family history of breast and ovarian cancer (among first-degree female relatives) was lower than NHW participants (*p <* 0.05). Interventions following *BRCA* genetic testing differed significantly between groups. Risk-reducing surgery was higher in NHW participants and using risk-reducing medication(s) was higher in BIPOC participants. The NHW participants were more likely to communicate genetic test results with healthcare providers, family, and friends than BIPOC participants (*p <* 0.05).

### 3.3. Participant Interactions with Healthcare, Coping Response, and Patient Activation

Healthcare interactions, aspects relating to coping response, and patient activation are depicted in Table 3. The NHW and BIPOC participants were similar regarding decision regret, general racism in the healthcare system, and MICRA sub-domains of distress and positive experiences. BIPOC participants had lower satisfaction with testing decisions and significantly higher ratings of personal discrimination (both single and multiple item measurements), fatalism scores (all dimensions), resilience scores, MICRA uncertainty, and lower patient activation (PAM) scores (*p <* 0.05).

### 3.4. Predictors of Patient Activation (Intention) and Communicating Test Results (Behavior)

We observed a significant relationship between patient activation (i.e., behavioral intention) and communication of test results with family members (i.e., behavior) among participants (*p* = 0.032). Test result (positive, negative, or variant of uncertain significance) was not associated with communication of risk with a family member (*p* = 0.217). Participants with higher activation scores were more likely to share their genetic test results with family members. We built a structural equation model (SEM) based on the TPB and included variables that differed significantly between NHW and BIPOC individuals. We used SEM to explore the effects of behavioral beliefs (i.e., satisfaction with genetic counseling and testing decision), normative beliefs (i.e., race/ethnicity, education, personal experiences of discrimination, health literacy, the importance of religion), control beliefs (i.e., fatalism, resilience, and MICRA uncertainty), and behavioral intention (i.e., patient activation) on behavior (i.e., communication of test results with family members) (Table 4). Assessing the direct effect of variables on patient activation score revealed three variables that significantly affected patient activation (*p <* 0.05). Participants with higher satisfaction with testing decisions, lower fatalism, and higher resilience were more likely to have higher activation scores. When assessing the indirect effect, we considered the mediating effect of patient activation score (PAM). Three variables (education, satisfaction with decision, and resilience) significantly affected communicating *BRCA* test results with family members (*p <* 0.05). Participants with higher education levels, greater satisfaction with their testing decision, and lower resilience are more likely to communicate their *BRCA* test results with family members.

#### 3.4.1. Qualitative Findings Related to the Theory of Planned Behavior “Behavioral Beliefs”

Satisfaction with genetic testing relates to the behavioral belief that genetic testing is “good” and/or useful. Satisfaction with the testing decision (SWD) was significantly lower among BIPOC survey participants compared to NHW counterparts. Satisfaction with testing decisions significantly affected intention (PAM) and behavior (i.e., communicating test results with blood relatives). Interviewee P022 (48-year-old Black female) had a low SWD score (vs. group mean of 4.3 ± 0.7). She was dissatisfied with her testing decision because she initially believed inheritance could only be passed through maternal lineage. P022′s PAM score was comparatively lower at 3.3 and she only communicated *BRCA*-related cancer risk to first-degree family members. In contrast, the individual with the highest SWD score (P019: 38-year-old NHW female) of five expressed no regret with her testing and preventative surgery decisions. This also shows in her higher PAM score of 3.8 and her willingness to communicate with all members of her family (Table 5).

#### 3.4.2. Qualitative Findings Related to the Theory of Planned Behavior “Normative Beliefs”

The TPB normative beliefs relate to an individual’s perceived norms, such as expectations that affect how an individual navigates their healthcare. The survey revealed education level as a significant factor in determining how individuals navigate a complex healthcare ecosystem. Findings from structural equation modeling (Table 4) reveal that health literacy was not directly related to patient activation, suggesting that decision-making for genetic testing is complex and extends beyond mere comprehension. A 28-year-old female (P003) identifies as mixed race (Caucasian, Black, and Hispanic) and holds an associate degree. She expressed feeling ‘rushed’ and frequently ‘ignored’ when discussing the emotional impact of genetic testing (i.e., how and when to inform her family of *BRCA* cancer risk). In contrast, a 65-year-old NHW female with an advanced degree (P005) shared her perception that health systems as a whole do not make it easy to access information or discuss more personal and emotional topics. However, she emphasized that she was fortunate to have the knowledge and resources to navigate the healthcare system in a way that was ‘best’ for her. She communicated that a combination of her personality, formal education, and health literacy allowed her to make the system work “in her favor” (Table 5).

#### 3.4.3. Qualitative Findings Related to the Theory of Planned Behavior “Behavioral Control Beliefs”

Behavioral control refers to an individual’s sense of agency and self-efficacy. Compared to NHW participants, BIPOC participants had significantly higher fatalism scores, which significantly and negatively affected patient activation (PAM). For example, P007 did not exhibit pre-conceived notions of having “bad” or “good” luck, nor did she believe her health was destined because of her genetic results. Her trust in science and the medical field allowed her to dispel any of these notions and believe she had control over her health, allowing her to have a higher patient activation measure score (4.33). Of course, trust in the medical system is not a luxury held by all, especially by those who have experienced racial discrimination and have been mistreated. P003, a 28-year-old “Mixed” female, for example, has had ongoing health issues all her life, and her experiences with IVF treatments due to her *BRCA* testing have reinforced her beliefs of “bad luck” and have lowered her PAM score (3.25).

Compared to NHW participants, BIPOC participants had significantly higher scores for resilience. Resilience significantly affects patient activation (PAM) and communication with blood relatives. Interestingly, those with higher resilience scores were less likely to share genetic test results with blood relatives. Resilience was a common theme in many interviews, and inter-individual differences could be seen in lived experience. For example, a 25-year-old NHW woman (P020) shared experiences of having to ‘fight’ for care and how difficult she finds it to self-advocate—due to the stress she feels upon entering medical facilities. In contrast, P011 is a 35-year-old NHB woman who scored highly on resilience. She described her emotionally challenging experience advocating for herself and other Black women when receiving care but emphasized the importance of resilience to protect herself and others (Table 5).

#### 3.4.4. Qualitative Findings Related to the Theory of Planned Behavior “Behavioral Intention”

In the Theory of Planned Behavior, behavioral intention lies between beliefs and actual behaviors. In this study, Patient Activation Measure (PAM) was used to quantify this intention. BIPOC participants had significantly lower PAM scores compared to NHW participants, reinforcing previous literature investigating disparities in the medical system and consequent distrust of the system and medical professionals by BIPOC individuals. For example, P009, a 30-year-old Black or African American Female, expressed she did not feel supported nor believed by her oncologist, making her less inclined to go to appointments and continue surveillance. This is reflected by her PAM score of 3.42. In contrast, P005, a 65-year-old Caucasian female, detailed how she had immediate access to care and was able to undergo surgery in a matter of weeks with a higher PAM score of 4 (Table 5).

#### 3.4.5. Qualitative Findings Related to the Theory of Planned Behavior “Behavior”

In the results of the quantitative survey, BIPOC participants had a significantly lower rate of communicating test results to their family members compared to NHW participants. In the case of our interview sample, only one outlier had no communication with their family. As a result, we measured the level of communication of interview participants by whether or not they communicated beyond their first-degree family members. For example, P011, a 35-year-old Black or African American female, had limited communication with their family members because they were worried about being a burden and did not tell anyone for ten years. In contrast, P004, a 65-year-old Mexican American female, was explicit in her hopes to share all information with extended family and even detailed her plans to educate her future grandchildren when they became of age (Table 5).

### 3.5. Expanding Survey Findings with Qualitative Interviews

In total, 18 individuals were invited to participate in the in-depth qualitative interviews. We used purposive sampling to reflect a range of sociodemographic characteristics among participants (Table 1). Table 5 depicts quantitative and qualitative findings using a “joint display” [29], directly linking significant quantitative survey findings with qualitative findings.

## 4. Discussion

Genetic testing for *BRCA* and cascade carrier screening can inform cancer risk reduction strategies, enable earlier intervention, and improve cancer outcomes. Although genetic testing has increased, it remains underutilized as only ~20% of high-risk breast cancer patients and ≤10% of asymptomatic individuals with a significant family cancer history receive testing [36]. Further, underutilization of genomic healthcare is most striking among individuals identifying as BIPOC. Many factors affect an individual’s decision to undergo genetic testing, including the history of a known pathogenic variant in the family, patterns of family history of cancer, insurance coverage, family planning considerations, and the psychological impact of a test result [3,12,37,38]. Consequently, genetic healthcare services ideally involve genetic healthcare providers as well as other specialists such as mental health and reproductive health providers [38].

The present study examined multi-level factors affecting intrafamilial communication of *BRCA* risk. We did not specifically interrogate specific individual factors such as reproductive status (e.g., having or planning to have children) and reproductive decision-making (e.g., decisions regarding risk-reducing surgery) that may affect risk communication. A recent mixed methods study of *BRCA*+ women analyzed responses from 505 *BRCA*+ participants and 40 in-depth interviews [39]. Investigators observed that age had a significant effect on reproductive decision-making as very few (6%) of reproductive age women had children after learning their *BRCA* status. Younger women were significantly more likely to have a child after testing—an event that was considered to be an important life milestone (i.e., childbearing/breastfeeding). Moreover, investigators found that older women (>40 years.) and women who had children prior to their *BRCA*+ status were more focused on cancer risk in family members and children compared to younger and childless counterparts. In this study, the BIPOC group was younger (35.1 ± 12.2) than the NHW counterparts (44.1 ± 13.0). The observed age difference may help explain the observation that the BIPOC individuals were more likely to opt for risk-reducing medication and less risk-reducing surgery compared to NHW individuals. Thus, age and reproductive status appear to be critical factors for decision-making and intrafamilial communication of risk. However, studies reported on a predominantly NHW population, so further work is needed to elucidate the role of reproductive status/decision-making for BIPOC individuals.

In this study, we employed targeted recruitment strategies to enhance the recruitment of BIPOC participants and better understand their experiences with genomic healthcare (i.e., genetic testing for *BRCA*), patient activation (intention), and intrafamilial communication of risk (i.e., behavior). A recent study examining family members of high-risk cancer individuals (e.g., harboring pathogenic *BRCA* variants) found lower rates of genetic counseling and testing among racial/ethnic minorities [6]. In our sample, we observed BIPOC participants had less family history of cancer—yet, a greater personal history of cancer and underwent *BRCA* testing at a younger age than NHW counterparts. BIPOC individuals exhibited higher scores on personal discrimination than NWH participants—indicating they had experienced discrimination while receiving healthcare services. Being a ‘minority’ or ‘minoritized’ in a community may affect an individual’s experience with the healthcare system. Health outcomes vary widely by race and ethnicity both across and within states. However, healthcare performance is low even in the states with the country’s largest Asian American, Native Hawaiian, and Pacific Islander populations [40]. We did not consider the participant’s geographic location in the present study was not investigated. It is plausible that discrimination might be felt or experienced more prominently in states/communities where BIPOC individuals are a minority compared to majority-BIPOC locations. Further, it would be important to know if a strong sense of community acceptance in BIPOC-majority cities influences patient activation. As such, it would be useful for future studies to consider the geographic location, healthcare system characteristics, and healthcare policies that may affect BIPOC individuals’ experiences with healthcare systems. BIPOC participants also had higher fatalism scores compared to their NHW individuals. Fatalism comprises predetermination (a belief that one’s health condition/experience is decided before they actually occur), luck (health is a matter of chance), and pessimism (the worst will happen) [27]. While our study design precludes identifying a causal link between personal experiences of discrimination and fatalism, it is plausible that experiences of discrimination could contribute to a perceived lack of control of one’s medical situation (i.e., predetermination, luck, and pessimism). Such structural impact aligns with findings from a recent study identifying distrust in healthcare services as a key barrier contributing to lower genetic testing rates among racial/ethnic minorities [6].

In the present study, BIPOC participants exhibited significantly lower activation (intention) as measured by the PAM. Consistent with the guiding theoretical framework (TPB), communicating *BRCA* risk to family members was lower among BIPOC participants (79.3%) compared to NHW (96.4%). Overall, we observed that individuals with higher activation were more likely to share genetic test results with family members. Such observations support the validity and utility of the TPB as a framework for understanding intrafamilial communication of risk—a prerequisite for cascade screening and subsequent risk-reducing measures to improve cancer outcomes. Our findings are consistent with prior work indicating lower rates of risk communication among young, *BRCA*+ Black women compared to those with a negative result (or a variant of uncertain significance) [16,41]. Considering all variables in the present study, test results (positive vs. negative) and self-identifying as BIPOC did not affect risk communication with family members. This observation suggests that decision-making surrounding intrafamilial communication of *BRCA* risk is complex. Those individuals with a higher education level, greater satisfaction with their testing decision, and lower resilience were more likely to communicate test results with family members.

Prior work has identified that educational attainment affects one’s understanding of test results and risk perceptions [17,42,43,44]. Importantly, evidence suggests that having higher levels of education does not ensure that an individual has higher levels of health literacy/numeracy [42]. We identified that greater satisfaction with the genetic testing decision (SWD) increased the likelihood of intrafamilial communication of *BRCA* risk—consistent with others’ findings highlighting the importance of knowledge and SWD with communication of risk [17,45]. In this study, BIPOC participants exhibited similar decision regret compared to NHW individuals—yet, lower satisfaction with their testing decision and higher uncertainty regarding testing results. One possible explanation for this difference could be a lack of racial/ethnic concordance between patients and genetic counselors. Data indicate that genetic counselors are predominantly NHW females and may be susceptible to implicit bias [46]. Further, a recent scoping review identified a lack of racial/ethnic concordance as a barrier to the uptake of prenatal genetic testing among Hispanic/Latinx individuals [47]. Thus, an important intervention at the healthcare system level is to diversify the racial/ethnic composition of providers of genomic healthcare. Providers need to be able to provide culturally empowered counseling and support for those with unique experiences in a healthcare setting due to their race and ethnicity (Figure 3). Such capacity building will take time, so more immediate interventions are warranted for pre-/post-test genetic counseling and support. Our findings underscore the importance of person-centered decisional support to empower individuals considering genetic testing by providing information and opportunity for reflection to support high-quality decisions that are informed and aligned with one’s values and preferences, as well as tailored support to foster disclosure of testing results to potentially at-risk blood relatives (Figure 3).

A somewhat surprising finding was that individuals with greater resilience scores were less likely to disclose *BRCA* risk to family members. Resilience refers to the process of sustaining physiological/behavioral stability in response to a stressful life event [48]. A cancer diagnosis and undergoing cancer treatment are associated with significant psychosocial distress [49]. Similarly, learning *BRCA*+ status is often a watershed moment accompanied by significant uncertainty and emotional distress [14]. We used qualitative interviews to explore the inverse relationship between resilience and intrafamilial communication of risk. Interviews revealed stories indicating that individuals with lower resilience often seek support from family members—thus creating opportunities for test results to be divulged. In contrast, interviews with individuals with higher resilience scores (Table 5) revealed a desire to remain private and ‘go it alone’. Resilience is a dynamic adaptation process that may change over time and can be increased by making meaning of one’s situation [49]. In addition to resilience, other factors that affect intrafamilial communication of *BRCA* risk, including vulnerability, emotional/geographic distance, and familial cohesion, mediate intrafamilial risk communication [14,43]. Our findings suggest that tailored support for individuals with greater resilience may be important for encouraging and ‘nudging’ individuals to share test results with at-risk blood relatives to enhance cascade screening efforts and improve cancer outcomes (Figure 3).

### Strengths and Limitations

This mixed-methods study included a relatively diverse, sizeable study sample recruited through *BRCA* community partnerships with BIPOC patient support organizations. Linking quantitative and qualitative data on the same individual allowed us to gain deeper explanatory insights into quantitative survey findings. The present data provide empirical support for the utility and validity of the Theory of Planned Behavior in understanding patient activation (i.e., intention) and intrafamilial communication of risk (i.e., behavior). As such, this framework may be valuable for guiding intervention development to surmount barriers to communicating *BRCA* risk and supporting subsequent cascade carrier screening to improve cancer outcomes. There are several limitations to this study. First, male participants were excluded so findings only apply to females. Little is known about BIPOC males’ experiences with genomic healthcare, patient activation, and intrafamilial risk communication and this gap merits attention. Similarly, few subjects from gender minorities (e.g., transgender and nonbinary) were included, so caution is warranted when extrapolating findings to these specific patient populations. Despite targeted recruitment strategies that included partnering with patient advocacy groups, we did not attain a sample that fully represents the general U.S. population. Further, we did not consider the race/ethnic composition of the geographic location of participants where BIPOC individuals may have different experiences with discrimination, racism, and the health care system. Mistrust of healthcare and research among BIPOC communities has been well-established [50,51]. We suspect that longstanding mistrust may have contributed to our falling short of a truly representative sample. Such experiences highlight the need for culturally empowered approaches and significant, ongoing time investment to cultivate trusting relationships via collaborative engagement with “cultural brokers”, advocacy groups, and other community-based organizations to build trust within BIPOC communities and enhance research participation. Further, most qualitative interviews were conducted with individuals who had shared their *BRCA* status with family members. As such, future work should continue to focus on diverse groups regarding race, ethnicity, sex, and gender identity.

## 5. Conclusions

We identify significant differences in experiences with genetic health services, patient activation, and intrafamilial risk communication between NHW and BIPOC individuals. Compared to NHW counterparts, BIPOC participants experienced more discrimination in healthcare, held a more fatalistic view of their condition, and had lower rates of asymptomatic testing as well as lower patient activation (i.e., knowledge, skills, and confidence in self-managing their condition). Guided by the TPB, we found that intrafamilial communication of risk was more likely among those with greater educational attainment, higher satisfaction with genetic testing decisions, and lower resilience scores. Disparities in genomic healthcare for *BRCA*-tested BIPOC individuals are complex and multifaceted. Bridging disparities to ensure that genomic healthcare benefits all people may demand theory-driven, multi-level interventions targeting the individual (education), interpersonal (resilience), and healthcare system (satisfaction with a testing decision) levels to enhance cascade carrier screening and improved cancer outcomes. At the health system level, we need to effectively recruit, train, and retain diverse healthcare providers in genomic healthcare. A representative workforce is particularly important for BIPOC individuals who are more likely to experience personal discrimination and be less satisfied with genetic counseling. At the interpersonal level, better understanding and consideration of unique circumstances affecting intrafamilial risk communication is needed. Specifically, individuals with higher levels of resilience (likely due to lived experiences) appear to need support for communicating test results to family members—who might not be receptive to such discussions. At the individual level, person-centered support is needed to increase informed decision-making that reflects the values and preferences of individuals—thereby bolstering satisfaction with testing decisions. This is specifically important for BIPOC individuals who exhibited lower satisfaction with testing decisions compared to the NHW counterparts.

## Figures and Tables

**Figure 1 genes-14-01450-f001:**
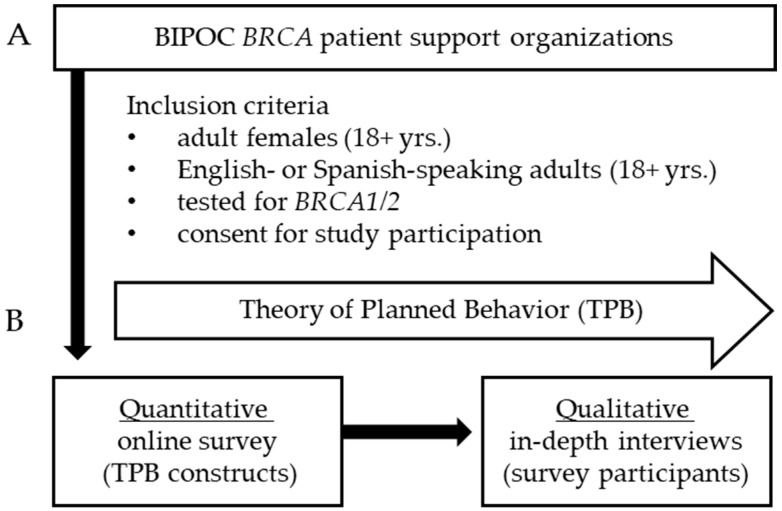
Study schematic. (**A**) Study recruitment was conducted in collaboration with BIPOC *BRCA* patient organizations to identify individuals meeting study criteria. (**B**) Guided by the Theory of Planned Behavior (TPB), a quantitative online survey assessing TPB constructs (Figure 2), was followed by a sequential qualitative arm with in-depth interviews to gain a deeper understanding of significant survey findings. Quantitative and qualitative data streams were then merged to identify targets for tailored multi-level interventions (i.e., at the individual, interpersonal, and health system levels) to enhance intrafamilial communication of *BRCA* risk.

**Figure 2 genes-14-01450-f002:**
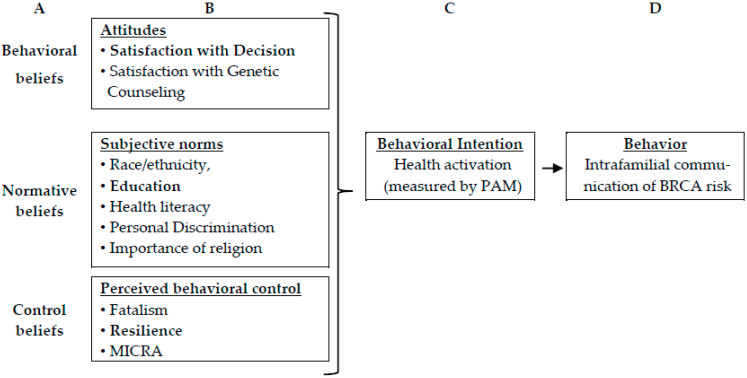
Theory of Planned Behavior. (**A**) Beliefs shape (**B**) individual attitudes (positive vs. negative), perceptions of peer/family/societal norms and expectations, and one’s perceived ability and autonomy for making a decision. These elements drive (**C**) intention (activation) and (**D**) subsequent behavior (risk communication). Abbreviations: PAM, Patient Activation measure, MICRA, Multidimensional Impact of Cancer Risk Assessment.

**Figure 3 genes-14-01450-f003:**
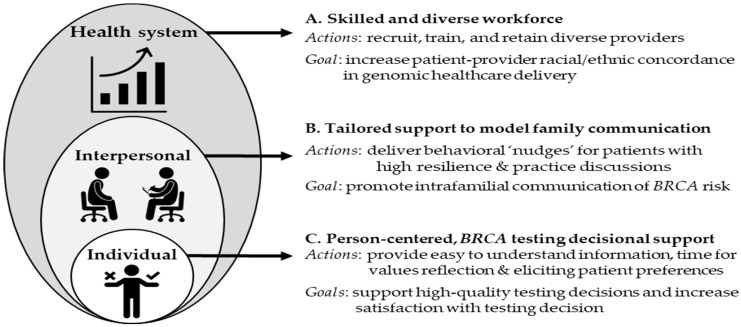
Systems perspective of proposed interventions to increase family communication of *BRCA* risk. (**A**) Health system-level interventions aim to build genomic healthcare workforce capacity to match patient diversity and enhance patient–provider concordance. (**B**) Interpersonal level interventions aim to deliver tailored interventions to increase motivation and confidence for communicating *BRCA* risk to blood relatives and enhance cascade carrier screening. (**C**) Individual-level interventions aim to promote culturally empowered pre-test genetic counseling supporting high-quality decisions and resulting satisfaction with a decision—a key predictor of communicating *BRCA* risk.

**Table 1 genes-14-01450-t001:** Socio-demographic of participants by race/ethnicity (n = 242).

Characteristics	Survey Participants			Interview Participants (n = 18)
	NHW (n = 197)	BIPOC (n = 45)	All (n = 242)	
	*n* (%)	*n* (%)	*n* (%)	
**Age (years)**				
min.–max.	20–76	20–70	20–76	25–68
mean ± S.D.	44.1 ± 13.0	35.1 ± 12.2 *	42.4 ± 13.3	41.8 ± 13.5
**Gender (current)**				
female (cis)	193 (98%)	28 (96.6%)	221 (97.8%)	18 (100%)
male	1 (0.5)	-	1 (0.4%)	-
transgender male	-	1 (3.4%)	1 (0.4%)	-
nonbinary/queer	3 (1.5%)	-	3 (1.3%)	-
**Marital status**				
married/in a relationship	155 (78.7%)	20 (69%)	175 (77.4%)	13 (72.2%)
single	24 (12.2%)	7 (24.1%)	31 (13.7%)	3 (16.7%)
separated/widowed	18 (9.1%)	2 (6.9%)	20 (8.8%)	2 (11.1%)
**Education**				
high school	6 (3%)	3 (10.3%) *	9 (4%)	-
some college/Associate’s degree	34 (17.3%)	3 (10.3%) *	37 (16.4%)	2 (11.1%)
college or advanced degree	157 (79.7%)	23 (79.3%)	180 (79.6%)	16 (88.9%)
**Health literacy** (scale range: 0–5; low is better)	1.24 ± 0.52	1.59 ± 0.81 *	−3.145/0.002	1.31 ± 0.63
**Household income (annual)**				
<$75,000/year.	106 (56.4%)	12 (41.4%)	118 (54.4%)	7 (38.9%)
$75,000–125,000/year	52 (27.7%)	13 (44.8%)	65 (30%)	6 (33.3%)
>$125,000/year	30 (16%)	4 (13.8%)	34 (15.7%)	5 (27.8%)
**Children**				
number of children (range: 1–4)	1.53 ± 0.67	1.98 ± 0.80 *	1.89 ± 0.79	2.33 ± 1.11
biological child(ren)	112 (57.4%)	21 (72.4%)	133 (59.4%)	5 (27.8%)
adopted child(ren)	14 (7.2%)	1 (3.4%)	15 (6.7%)	2 (11.1%)
both biological and adopted children	6 (3.1%)	-	6 (2.7%)	2 (11.1%)
no children	63 (32.3%)	7 (24.1%)	70 (31.3%)	9 (50%)
**Importance of religion (0–10)**	6.04 ± 2.38	4.58 ± 3.37 *	4.88 ± 3.24	5.52 ± 3.90

* *p <* 0.05 between BIPOC and NHW participants.

**Table 2 genes-14-01450-t002:** Cancer history, genetic testing, and medical decision-making (n = 242).

	Survey Participants		
Characteristics	NHW (n = 197)	BIPOC (n = 45)	All (n = 242)
	*n* (%)	*n* (%)	*n* (%)
**Personal history of cancer**	80 (40.6%)	18 (62.1%) *	98 (43.4%)
Breast/ovarian cancer in 1st female relatives	87 (44.2%)	6 (20.7%) *	93 (41.2%)
Breast cancer in 1st male relatives	21 (10.9%)	12 (41.4%) *	33 (14.9%)
Age at *BRCA* genetic testing (years)			
min.–max.	18–73	18–68	18–73
mean ± S.D.	38.8 ± 12.53	31.0 ± 12.26*	37.5 ± 12.80
**Pathogenic *BRCA* variant**			
yes	179 (90.9%)	25 (86.2%)	204 (90.3%)
no	15 (7.6%)	4 (13.8%)	19 (8.4%)
uncertain	3 (1.5%)	-	3 (1.3%)
**Inheritance of *BRCA* variant**			
maternal	93 (52%)	14 (56%)	107 (52.5%)
paternal	59 (33%)	8 (32%)	67 (32.8%)
both	2 (1.1%)	2 (8%)	4 (2%)
unknown	25 (14%)	1 (4%)	26 (12.7%)
**Medical decisions after *BRCA* testing**			
increased surveillance	42 (23.5%)	11 (29.7%)	54 (21.9%)
risk-reducing medication	6 (3.4%)	16 (43.2%) *	22 (9.1%)
risk-reducing surgery	111 (62.0%)	9 (24.3%) *	120 (49.6%)
combination of surveillance/risk-reducing surgery	7 (3.9%)	1 (2.7%)	12 (5%)
none	2 (1.1%)	-	2 (0.8%)
Other ^†^	11 (6.1)	-	7 (2.9%)
**Primary reason for genetic testing**			
personal desire for information	74 (37.6%)	11 (37.9%)	85 (37.6%)
provider suggestion/encouragement	52 (26.4%)	12 (41.4%)	64 (28.5%)
family suggestion/encouragement	50 (25.4%)	4 (13.8%)	54 (23.9%)
other (suggestions from others)	21 (10.7%)	2 (6.9%)	23 (10.2%)
**Communicated *BRCA* risk with:**			
healthcare provider(s)	161 (81.7%)	13 (44.8%) *	174 (77%)
family members/blood relatives	190 (96.4%)	23 (79.3%) *	213 (94.2%)
friends	152 (77.2%)	12 (42.9%) *	164 (72.9%)

* *p <* 0.05 between BIPOC and NHW participants; ^†^ Currently undecided weighing increased surveillance or/and risk-reducing surgery.

**Table 3 genes-14-01450-t003:** Measures assessing healthcare interactions, coping response, and patient activation (n = 242).

	NHW Participants	BIPOC Participants	z Value/*p* Value	All Participants
**Genetic Counseling Satisfaction Scale** (scale range: 0–100)	81.5 ±17.8	71.8 ± 22.8 *	−2.658/**0.008**	79.5 ± 19.3
**Satisfaction with Testing Decision** (scale range: 1–5)	4.4 ± 0.6	3.80±.95 *	−4.116/**<0.001**	4.3 ± 0.7
**Decision Regret Scale** (scale range: 0–100)	51.2 ±5.6	49.68±10.90	−.390/0.697	50.9 ± 6.9
**Personal Discrimination** Single-Item (scale range: 0–1)	0.04 ± 0.18	0.73 ± 0.44 *	−11.34/**<0.001**	0.17 ± 0.37
Multi-item (scale range: 0–7)	4.0 ± 0.9	5.4 ± 1.7 *	−2.193/**0.028**	5.2 ± 1.6
**General Racism in Healthcare System** (scale range: 0–4)	2.0 ± 1.7	1.9 ± 1.3	−0.073/0.942	1.9 ± 1.4
**Fatalism scale** (scale range: 0–5)	2.2 ± 0.6	2.7 ± 0.6 *	−2.319/**0.020**	2.2 ± 0.6
Predetermination	2.2 ± 0.7	2.9 ± 0.6 *	−3.138/**0.002**	2.3 ± 0.7
Luck	2.3 ± 0.9	2.9 ± 1.1 *	−2.993/**0.003**	2.40 ± 0.99
Pessimism	2.2 ± 0.8	3.0 ± 1.0 *	−5.021/**<0.001**	2.35 ± 0.86
**MICRA**(scale range: 0–5)	2.0 ± 0.8	2.3 ± 0.8 *	−2.084/**0.037**	2.06 ± 0.78
Distress	2.1 ± 1.2	2.4 ± 1.3	−1.630/0.103	2.16 ± 1.20
Uncertainty	1.7 ± 0.8	2.0 ± 0.9 *	−2.204/**0.027**	1.73 ± 0.85
Positive experience	2.2 ± 1.0	2.0 ± 1.2	−1.369/0.171	2.13 ± 1.07
**The Brief Resilience Scale** (scale range: 0–6)	3.0 ± 0.3	3.2 ± 0.7 *	−2.767/**0.006**	3.01 ± 0.39
**The Patient Activation Measure** (scale range: 0–100)	3.5 ± 0.5	3.2 ± 0.6 *	−3.039/**0.002**	3.43 ± 0.53

* *p* < 0.05 between BIPOC and NHW participants; MICRA: Multidimensional Impact of Cancer Risk Assessment.

**Table 4 genes-14-01450-t004:** Predictors of patient activation and communication of *BRCA* test results with family members.

Variables	Coefficient	S.D.	z	*p* Value
**Direct effects of variables on patient activation score**				
race/ethnicity (BIPOC vs. NHW)	−0.0189	0.0868	−0.22	0.827
education (B.S. vs. non-B.S. degree)	0.0362	0.0707	0.51	0.608
health Literacy	−0.1077	0.0642	−1.67	0.094
importance of religion	0.0023	0.0093	0.25	0.802
Personal Discrimination Scale	−0.0049	0.1038	−0.05	0.962
Genetic Counseling Satisfaction Scale	0.0027	0.0018	1.44	0.149
Satisfaction with Decision Scale	0.3239	0.0559	5.79	**<0.001**
Fatalism Scale	−0.2481	0.0771	−3.21	**0.001**
Brief Resilience Scale	0.2495	0.0779	3.20	**0.001**
MICRA (uncertainty)	−0.0084	0.0372	−0.23	0.821
**Indirect effect of variables (with mediating effect of patient activation score) on communicating results**				
Patient Activation Measure	0.0347	0.0581	0.60	0.550
race/ethnicity	−0.0772	0.0754	−1.02	0.306
education	−10.102	0.0484	−2.11	**0.035**
health literacy	−0.0092	0.0337	−0.27	0.785
importance of religion	0.0034	0.0040	0.86	0.388
Personal Discrimination Scale	−0.1057	0.0868	−1.22	0.223
Genetic Counseling Satisfaction Scale	0.0009	0.0012	0.73	0.463
Satisfaction with Decision Scale	0.0768	0.0344	2.23	**0.026**
Fatalism Scale	0.0878	0.0583	1.51	0.132
Brief Resilience Scale	−0.1459	0.0608	−2.40	**0.017**
MICRA (uncertainty)	0.0021	0.0247	0.09	0.930

MICRA: Multidimensional impact of cancer risk assessment.

**Table 5 genes-14-01450-t005:** Joint display of quantitative and qualitative findings.

Significant Quantitative Findings	Representative Qualitative Interview Quotes (Providing Context and Insight into Quantitative Findings)
**Behavioral beliefs** Satisfaction with Decision (SWD): BIPOC: 3.8 ± 1.0 NHW: 4.4 ± 0.6 (*p <* 0.001)	**Low Satisfaction** P022: 48 y.o. female, Black/African American, SWD: 4.3/5, PAM: 3.3/5 communicated risk to 1st degree family members. *“Because in spite of what I believed, even up to the moment I was diagnosed with the first lump, I still… I still… that day, I was like, [...] why am I going to get this stupid mammogram because it comes back negative in areas? The waste of my time, and it’s painful, and it’s uncomfortable, and I have a meeting to get to… I don’t want to be late for my meeting, you know? So even up until that point in time, I still didn’t believe that I could get breast cancer because of what I believe about… you know… my mother and my grandmothers not having it [cancer], not even thinking about my father’s side of the family.”* **High Satisfaction** P019, 38 y.o. female, NHW, SWD: 5.0/5, PAM: 3.8/5, communicated risk to family members. *“Sometimes it flares up in, like, weird situations. But um… but for the most part, I feel good. What I feel good about is the decisions that I made. I don’t regret them. And I feel like they were the right decisions. And I’m never going to live without any anxiety. So I think I did the best that I could, for, like, you know, being able to live the best that I can”*.
**Normative beliefs** Education level (non-B.S. degree) BIPOC: 20.6% NHW: 20.3% (*p <* 0.05 )	**Associate’s degree** P003: 28 y.o. female, mixed race health literacy score: 2/5, communicated risk to family members. *“At the time, I was a little bit naïve. Before that appointment, I believed I was going to be the same as I was then [...] Maybe just media, maybe just naïvete… this, idea that like… well, he’s a plastic surgeon, and he says it’s—you know… he does all of this stuff. Like, it’ll look the same—[after risk-reducing surgery]”* **Graduate degree** P005: 65 y.o. female, NHW health literacy score: 1/5, communicated risk to family members. *“I think that [education/knowledge] takes off a huge amount of pressure, and stress, and worry. I was able to do what I did, because I… basically, I knew how to access information. I know how to pick up the phone, I wasn’t afraid to pick up the phone [...] I mean, you know, it was a lot easier for me, because I’m very, I’m very privileged”*.
**Behavioral control** Total Fatalism Scale BIPOC: 2.68 ± 0.58 NHW: 2.20 ± 0.57 (*p* = 0.02)	**Low fatalism** P007: 57 y.o. female, NHW, Fatalism Score: 1.63/5, PAM: 4.33/5 *“I have the information [genetic test results] and you know… my degree is in science, so I come from that background and the importance… just the absolute importance [of the result]. And then, if you do… if you do get diagnosed, this is what you can do! You know, just don’t stick your head in the sand.”* **High fatalism** P003: 28 y.o. female, mixed race, Fatalism score: 3.04/5, PAM: 3.25/5 *“You know, and it’s…it’s been so hard because I… I acknowledge the fact that it’s vital to ensure that this doesn’t happen. But it… it’s such a… I just wish I could have kids the normal way. Yeah, because I feel like my whole life has just not… it’s never been easy. And, yeah, I think because of that, some genetic things, I… I don’t… I have always been sick”*.
Brief Resilience Scale BIPOC: 3.18 ± 0.65 NHW: 2.98 ± 0.30 (*p* = 0.006) higher resilience = less likely to communicate *BRCA* risk	**Low resilience** P020: 25 y.o. female, NHW, Resilience: 2.67/5, PAM: 3.25/5 Communicated risk to family members. *“When I walked in, the first person that came in to see me was a young female surgeon. And she just instantly… I have to laugh ‘cause they went in and I burst into tears. And I said, Look, I don’t know why I’m crying because I’m totally fine with it. But the minute I get into hospital, I just […] I just think like, I just clam up and it’s not… my head is perfectly fine. But for some reason, my face is just expelling tears.”* **High resilience** P011: 35 y.o. female, Black or African American, Resilience: 3.00/5, PAM: 4.00/5 Has a lower level of communicating risk to family. *“And it does, it takes a lot out of you, it is exhausting that I have to go to this level to get even the same service when you know, I feel like it to be equitable. But it’s exhausting. It’s tiring. It is disappointing. But at the same time, you can’t, you can’t leave it unchecked, like you have to address it. Because if you don’t, then you’re, you’re setting the expectation that it’s okay. And it’s going to be done to every other woman of color or Black woman that comes into the door behind me. And I can promise you she never did that again to another person who looks like me or another person of color”*.
**Behavioral Intention** Patient Activation Measure (PAM) BIPOC: 3.16 ± 0.60 NHW: 3.49 ± 0.50 (*p* = 0.002)	**Low PAM** P009: 30 y.o. female, Black/African American, PAM:3.42/5 *“Sometimes we [Black people] don’t get the best of care, or having people not really listen… having to really push, not having just all the information just given to us right away [...] I went back to the oncologist, and I was telling him what happened. It wasn’t common, so he didn’t hear this with anybody else. So it was kind of like… well, it could just have been the first time and that could also be symptoms of chemo… and it just wasn’t [chemotherapy effects]. He wasn’t… I felt like he didn’t believe me.”* **High PAM ** P005: 65 y.o. female, NHW, PAM: 4.00/5 *“I need referrals immediately. But in fact… what I did was… I just, I can’t stand not knowing. So I pay. And I’m very, very disapproving [of] private medicine. But, you know… in this instance, I just thought, sorry. I’m going for it. [...] I’m going to use my money. I saw somebody private, privately. And, I just… I just threw everything I had at it [...] And in fact, I was in hospital within about four weeks”.*
**Behavior** Intrafamilial communication of BRCA risk n = 213 (94.2%) BIPOC: n = 23 (79.3%) NHW: n = 190 (96.4%)	**Did not communicate *BRCA* risk to family** P011, Female, Black or African American, 35, 76–100k Communicated with first-degree family members only *“Oh, I definitely have the support-strong family support my dad supported me, but I only told him limited information because he was fighting his own journey, and I didn’t want him concerned about me especially since I knew that this disease- that this diagnosis didn’t mean a disease you know when I was diagnosed but then I also didn’t want to kind of worry him either and so I kind of just told him but I told him it’s you know, I’m the reassuring I don’t have cancer just means I’m high risk. And then I really chose to not tell other family members and friends only because, you know, I just I was felt, again -grieving and processing this all for myself, and so I didn’t really share with them and almost 10 years later, when I had my double mastectomy, and then at that time, over 10 years, I was fully educated and equipped to be able to share this news.”* **Communicated *BRCA* risk to family** P004: 65 y.o. female, Mexican-American Communicated with entire family *“I think I’m going to continue to be pretty proactive in terms of like, my own family. Making sure my siblings who haven’t been tested yet get tested. Making sure my own child or children are aware of their possible genetic mutations, you know? And, making sure that they make decisions in the future in terms of being proactive… if they want to. Hopefully, they want to, right? I think those are my… my main sort of my main outlook”*

## Data Availability

De-identified data will be made readily available upon request for research purposes to qualified individuals within the scientific community.

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
