# Peer review of "Racial and Ethnic Disparities in Genomic Healthcare Utilization, Patient Activation, and Intrafamilial Communication of Risk among Females Tested for BRCA Variants: A Mixed Methods Study"

_genes, 2023, doi:10.3390/genes14071450_

Round 1
Reviewer 1 Report
Title: “Racial and Ethnic Disparities in Genomic Healthcare 2 Utilization, Patient Activation, and Intrafamilial 3 Communication of Risk Among Females Tested for BRCA 4 Variants: A Mixed Methods Study” by Sharlene Hesse-Biber et al.
Simple Summary:
Decision making on intrafamilial communication of BReast CAncer gene (BRCA) 14 risk and subsequent cascade screening is complex and affected by individual, interpersonal, and 15 healthcare system-related factors. There is a paucity of evidence on factors affecting intrafamilial 16 risk communication, specifically among ethnically and racially diverse individuals tested for BRCA 17 variants. A deep understanding of multi-level factors affecting communication is needed to reap the 18 full potential of cascade screening for improving health outcomes. This study provides findings to 19 guide theory-driven, multi-level interventions to enhance the utilization of genomic healthcare for 20 diverse people.
The authors identified significant differences in experiences with genetic health services, patient activation, and intrafamilial risk communication between non-Hispanic white (NHW) and black, Indigenous, and people of color (BIPOC) individuals.
This reviewer thinks this manuscript is interesting and informative, however, some description need to be revised as follows.
Major points.
1. In table1, percentage of separated/widowed in NHW and BIPOC is confusing. Need to be corrected.
2. In table1, is household income per person or per family?
3. In table 2, there is big difference in medical decision after BRCA testing between NHW and BIPOC. Please describe the reason why the difference due to the education but not household income?
4. In figure3, how does authors think to promote individual step (C) to interpersonal step (B). Are there different approach of strategy between NHW and BIPOC?
5. In figure3, how does authors think to promote interpersonal step (B) to to health system (A). Is there different approach of strategy to accomplish individual step (C) to interpersonal step (B) or interpersonal step (B) to health system (A) between NHW and BIPOC? Please describe briefly in discussion.
Title: “Racial and Ethnic Disparities in Genomic Healthcare 2 Utilization, Patient Activation, and Intrafamilial 3 Communication of Risk Among Females Tested for BRCA 4 Variants: A Mixed Methods Study” by Sharlene Hesse-Biber et al.
Simple Summary:
Decision making on intrafamilial communication of BReast CAncer gene (BRCA) 14 risk and subsequent cascade screening is complex and affected by individual, interpersonal, and 15 healthcare system-related factors. There is a paucity of evidence on factors affecting intrafamilial 16 risk communication, specifically among ethnically and racially diverse individuals tested for BRCA 17 variants. A deep understanding of multi-level factors affecting communication is needed to reap the 18 full potential of cascade screening for improving health outcomes. This study provides findings to 19 guide theory-driven, multi-level interventions to enhance the utilization of genomic healthcare for 20 diverse people.
The authors identified significant differences in experiences with genetic health services, patient activation, and intrafamilial risk communication between non-Hispanic white (NHW) and black, Indigenous, and people of color (BIPOC) individuals.
This reviewer thinks this manuscript is interesting and informative, however, some description need to be revised as follows.
Major points.
1. In table1, percentage of separated/widowed in NHW and BIPOC is confusing. Need to be corrected.
2. In table1, is household income per person or per family?
3. In table 2, there is big difference in medical decision after BRCA testing between NHW and BIPOC. Please describe the reason why the difference due to the education but not household income?
4. In figure3, how does authors think to promote individual step (C) to interpersonal step (B). Are there different approach of strategy between NHW and BIPOC?
5. In figure3, how does authors think to promote interpersonal step (B) to to health system (A). Is there different approach of strategy to accomplish individual step (C) to interpersonal step (B) or interpersonal step (B) to health system (A) between NHW and BIPOC? Please describe briefly in discussion.
Author Response
Thank you so much for your feedbacks, we addressed the concerns raised by the reviewer.

Reviewer 2 Report
Hesse-Biber and colleagues present an interesting study employing a mixed-methods approach, discussing how utilization of healthcare resources differs between racial/ethnically different groups. They present evidence using mixed racial cohort of breast cancer patients carrying the BRCA mutant allele. While the study appears to have exploited the statistically relevant tools to produce calculated conclusions, one wonders about the far fewer number of BIPOC subjects included in the study compared to non-hispanic white subjects. Since the focus of the study is exceedingly on the thought process and actions of people of color, it would be scientifically expected to include at least an equal number of such persons if not more. As it stands, the study bases its findings on only 45 BIPOC subjects compared to 197 white subjects. Given the relatively poor cohort size of the persons most pertinent to this study, the use of statistics does not quite seem justified.
Furthermore, BIPOC by definition represents a several ethnic communities but seems to have been grouped as a single entity. The authors should clarify on the specific ethnicity, race of people included within the BIPOC group.
One aspect this study ignores is the geographical distribution of BIPOC and whether this group is a majority or minority in the region where the study was conducted. The arguments of the authors appear generalized which may not be realistic. In a minority-BIPOC city/state, discrimination might be felt/experienced more prominently than in majority-BIPOC locations. Are the healthcare concerns of BIPOC presented in this work, similar to that in BIPOC-dominated cities? Does a strong sense of community-acceptance in BIPOC-majority cities influence patient activation? This needs to be duly discussed by the authors in the manuscript, albeit with caution.
Author Response

(The authors gave the same response as above.)
